# Comparison of carbon and water fluxes and the drivers of ecosystem water use efficiency in a temperate rainforest and a peatland in southern South America

Jorge F. Perez-Quezada[1,2,3], David Trejo[1,2,3], Javier Lopatin[4,5,6], David Aguilera[1], Bruce Osborne[7,8], Mauricio Galleguillos[1,4,5], Luca Zattera[1], Juan L. Celis-Diez[2,9], and Juan J. Armesto[2,10]

[1]Department of Environmental Sciences and Renewable Natural Resources, University of Chile, Santiago, Chile.
[2]Institute of Ecology and Biodiversity, Barrio Universitario, Concepción, Chile.
[3]Cape Horn International Center, Punta Arenas, Chile.
[4]Faculty of Engineering and Science, Universidad Adolfo Ibáñez, Santiago, Chile.
[5]Centro de Ciencia del Clima y la Resiliencia, CR2, Santiago, Chile.
[6]Data Observatory Foundation, ANID Technology Center No. DO210001, Santiago, Chile.
[7]UCD School of Agriculture and Food Science, University College Dublin, Dublin 4, Ireland.
[8]UCD Earth Institute, University College Dublin, Dublin 4, Ireland.
[9]Escuela de Agronomía, Pontificia Universidad Católica de Valparaíso, Quillota, Chile.
[10]Departamento de Ecología, Pontificia Universidad Católica de Chile, Santiago, Chile.

**Correspondence:** Jorge F. Perez-Quezada (jorgepq@uchile.cl)

**Abstract.** The variability and drivers of carbon and water fluxes and their relationship to ecosystem water use efficiency (WUE) in natural ecosystems of southern South America are still poorly understood. For eight years (2015-2022), we measured carbon dioxide net ecosystem exchange (NEE) and evapotranspiration (ET) using eddy covariance towers in a temperate rainforest and a peatland in southern Chile. NEE was partitioned into gross primary productivity (GPP) and ecosystem respiration (Reco), while ET was partitioned into evaporation (E) and transpiration (T) and used to estimate different expressions of ecosystem WUE. We then used the correlation between detrended time series and structural equation modeling to identify the main environmental drivers of WUE, GPP, ET, E and T. The results showed that the forest was a consistent carbon sink (-486 $\pm$ 23 g C m$^{-2}$ y$^{-1}$) while the peatland was, on average, a small source (33 $\pm$ 21 g C m$^{-2}$ y$^{-1}$). WUE is low in both ecosystems, and likely explained by the high annual precipitation in this region ($\sim$ 2100 mm). Only expressions of WUE that included atmospheric water demand showed seasonal variation. Variations in WUE were related more to changes in ET than to changes in GPP, while T remained relatively stable, accounting for around 47% of ET for most of the study period. For both ecosystems, E increased with higher global radiation, higher surface conductance and when the water table was closer to the surface. Higher values for E were also found with increased wind speeds in the forest and higher air temperatures in the peatland. The absence of a close relationship between ET and GPP is likely related to the dominance of plant species that either do not have stomata (i.e., mosses in the peatland or epiphytes in the forest) or have poor stomatal control (i.e., anisohydric tree species in the forest). The observed increase in potential ET in the last two decades and the projected drought in this region suggests that WUE could increase in these ecosystems, particularly in the forest, where stomatal control may be more significant.

Keywords: Inherent water use efficiency, intrinsic water use efficiency, underlying water use efficiency, net ecosystem exchange, ecosystem respiration, northern Patagonia.

**1 Introduction**

Climate change is currently affecting the functioning of ecosystems around the world. Of particular interest is how climate change will modify the fluxes of carbon and water because these are central to an understanding of ecosystem water balance and future primary productivity. Moreover, with projections indicating significant changes in water availability in many ecosystems globally (Caretta et al., 2023), the efficiency of the use of water in photosynthesis is likely to play a key role in future

vegetation productivity. One way of assessing this is by studying plant water use efficiency (WUE), defined as the carbon gain per unit of water lost (Chapin et al., 2011). At the leaf level, WUE is determined by the response of stomatal conductance to carbon dioxide ($CO_2$) and water vapor exchange via transpiration. At the ecosystem scale, WUE is determined by the carbon uptake by vegetation, and the water lost through transpiration (T) and evaporation (E) and reflects the functional coupling between the water and carbon cycles (Bacon, 2004).

The most common way to estimate ecosystem WUE is as the ratio between gross primary productivity (GPP) and evapotranspiration (ET) (Beer et al., 2009), which can be obtained from eddy covariance observations (Brümmer et al., 2012). The eddy covariance method, although unable to resolve species-specific leaf or tree scale dynamics (Keenan et al., 2013), is particularly effective for coupling high temporal resolution WUE and meteorological data, allowing a better understanding of the environmental controls on ecosystem carbon and water fluxes (Yi et al., 2019).

According to most studies, WUE has increased over the last two decades, which is partially explained by (1) an increase in GPP due to rising atmospheric $CO_2$ concentration, which results in a higher net carbon gain, with or without a reduction in stomatal conductance and reduced transpiration rates (Keenan et al., 2013), and/or (2) a reduction in stomatal conductance caused by water-deficits (Saurer et al., 2004), which reduces transpiration to a greater extent than carbon assimilation. Nevertheless, other research has suggested that ecosystem WUE may decrease when climate warming (Boeck et al., 2006) or nitrogen deposition

are considered (Huang et al., 2015), and this response may vary depending on the ecosystem type and hydroclimate (Terán et al., 2023).

These contradictory hypotheses were assessed by Lavergne et al. (2019), who suggested improving long-term observation-based estimates of WUE and the use of different formulations of WUE. Besides the basic form of WUE (GPP/ET), other formulations include the effect of vapor pressure deficit (Beer et al., 2009) or the bulk surface conductance of the ecosystem

(Lloyd et al., 2002), which can increase our understanding of the physiological processes involved in the exchange of carbon and water at the ecosystem level.

A study that included all European ecosystems showed that WUE is more related to changes in GPP in drier ecosystems, whereas it is more related to changes in ET in more humid environments (Terán et al., 2023). As ET is determined by both E and T, partitioning of these would allow us to differentiate between biological (T) and physical (E) drivers of evaporative losses

(Paul-Limoges et al., 2020). On a global scale, it has been suggested that T fluxes make the greatest contribution to ET in con-

tinental ecosystems (Jasechko et al., 2013). However, reports from temperate rainforest ecosystems showed that T represented about 55% of annual ET in an Eastern white pine forest (Ford et al., 2007) and 43% in a planted coniferous forest (Shimizu et al., 2015). Few studies have investigated the partitioning of ET in wetland ecosystems, with T representing 45.6% of ET in a *Sphagnum* fen ecosystem (Kim and Verma, 1996) (where T was associated with vascular plants and E with *Sphagnum*) and 43% (range 17-73%) in an Alpine meadow (Cui et al., 2020). These results indicate that E can be of a similar or somewhat higher magnitude than T in ecosystems subjected to high water availability.

Estimations of WUE from direct measurements in Southern Hemisphere biomes are underrepresented in flux monitoring networks (Pastorello et al., 2020). Southern South America is experiencing pronounced climate warming, and it is also expected that the intensity of heavy precipitation, droughts, and fires will intensify through this century, while mean wind speed and precipitation will decrease (Castellanos et al., 2022). This scenario can influence the tradeoff between carbon uptake and water loss by plants (Bréda et al., 2006) and significantly impact the terrestrial water and carbon cycles. Studies on the carbon fluxes of old-growth forests and a peatland in southern South America using the eddy covariance technique have shown a net loss of carbon in response to summer drought (Perez-Quezada et al., 2018, 2023; Valdés-Barrera et al., 2019). However, no reports are available for water fluxes, WUE or its drivers, which are crucial for a more comprehensive understanding of the carbon and water cycles in the region.

Using eight years of eddy covariance data from a temperate rainforest and an associated peatland in southern Chile, the objectives of this study were to: 1) analyze seasonal variability and annual values of GPP, ET, and the use of different expressions of WUE in both ecosystems, 2) examine the relation between GPP and ET with WUE, 3) assess the contribution of evaporation and transpiration to ET and WUE, and 4) identify the main environmental drivers of GPP, ET and its components.

## 2   Materials and Methods

### 2.1   Study area and sites

Experimental data were obtained from a temperate rainforest and an adjacent anthropogenic peatland at the Senda Darwin Biological Station (41°52' S, 73°39' W), the latter being formed after a forest fire >50 years ago (AmeriFlux sites CL-SDF and CL-SDP, respectively). The Station is located 15 km east of the city of Ancud, in the northern part of Chiloé Island, Chile (Fig. 1 a-c), in a rural landscape mosaic of pastures, shrublands, and forest patches, at 25 m above sea level and about 6 km from the coast. The climate is temperate with a strong oceanic influence (Beck et al., 2018), with a mean annual temperature of 9.7 °C and a mean annual precipitation of 2087 mm, with the driest period from December to March (Perez-Quezada et al., 2021a). Soils are classified as Placic Andosols, which are waterlogged volcanic ash soils located on flat fluvial-glacial terraces (CIREN, 2003).

The forest site is a 100-ha patch of North Patagonian broadleaved evergreen temperate rainforest, dominated by *Drimys winteri*, *Nothofagus nitida*, *Saxegothaea conspicua*, and *Tepualia stipularis*, with a canopy height of ∼25 m (Fig. 1d), a mean leaf area index (LAI) of 3.7 (range 2.5-5.5) and a mean canopy openness of 5.4% (range 1.6-12.9) that allows the growth of understory vegetation, epiphytes and vines (Perez-Quezada et al., 2021a). Soils are highly organic (∼40.2% C), with a low bulk density

($\sim$0.36 g cm$^{-3}$), and shallow with a placic largely impermeable layer at $\sim$52 cm, which results in frequent water-saturated conditions (Perez-Quezada et al., 2021a).

The anthropogenic peatland site has a total area of 16 ha, of which 5.4 ha are included within the boundary of the Biological Station and has been protected for 20 consecutive years, while the remaining part is within private property and used for grazing. The peatland ecosystem originated through the flooding of the soil after the removal of trees due to the burning of the rainforest (Díaz et al., 2007). The area was colonized mainly by the moss *Sphagnum magellanicum*, which occupies 60% of the above-ground vegetation (Díaz et al., 2007) (Fig. 1e). The dominant herbaceous species are *Sticherus cryptocarpus* and *Juncus procerus*. Shrubs represent roughly 20% of the species in the site, which is dominated by *Gaultheria mucronata*, *Baccharis patagonica*, and *Myrteola nummularia* (Cabezas et al., 2015). Canopy height ranges from 0.1 to 1 m and the peat layer is relatively shallow ($\sim$40 cm), with the impermeable placic layer restricting root growth to the upper soil horizon (Cabezas et al., 2015). The peatland is frequently waterlogged during the Austral winter (June to August) but can dry out intermittently during the summer (Bustamante-Sánchez et al., 2011).

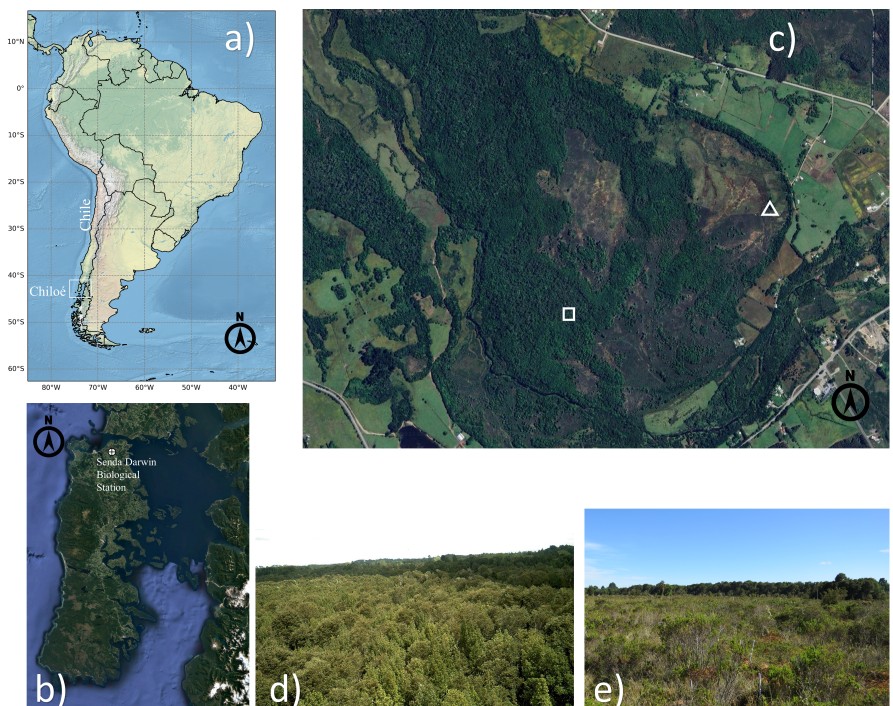

**Figure 1.** Location of the Senda Darwin Biological Station in northern Chiloé island (a, b). Highlighted in c) are the eddy covariance stations of the old-growth temperate rainforest (square) and the anthropogenic peatland (triangle). Views of the forest canopy (d) and the anthropogenic peatland (e). (a) Made with Natural Earth. (b, c) Map data ©2023 Google. (d, e) pictures taken by the author.

## 2.2 Meteorological measurements

The instruments installed at both sites adjacent to the EC stations recorded simultaneously the following micrometeorological and soil variables: net radiation ($R_n$), short-wave (SW) and long-wave (LW) components (NR01, Hukseflux, Delft, The Netherlands), photosynthetically active radiation (PAR; LI-190, LI-COR, Lincoln, Nebraska, USA), precipitation (P; 52202; RM Young, Traverse City, Michigan, USA), air temperature and relative humidity (Ta and RH; HMP155, Vaisala, Helsinki, Finland; hereafter Vaisala), soil temperature (Ts; TCAV thermocouples, CSI), volumetric soil water content (SWC; three sensors within 15 m of the towers using a water content reflectometer at 5 cm depth; CS616, CSI), water table depth (WTD; pressure transducers CS451, CSI), and wind speed measured at a height of 3 m in the peatland and 40 m in the forest (U; sonic anemometer CSAT3A, CSI). Data was recorded at 30-minute intervals (datalogger CR3000, CSI).

Additional meteorological variables covering both areas were available from the nearby Senda Darwin meteorological station for the period 1999-2021, which recorded solar radiation (Rg; LI-200S; LI-COR, Lincoln, Nebraska, USA), air temperature (Ta; HMP45A, Vaisala), and total precipitation (P; rain gauge, TR525M, Texas Electronics, Dallas, Texas, USA).

We computed daytime averages for Rg and Ta from the Senda Darwin meteorological station between 1999 and 2021 to derive annual variations in potential evapotranspiration (PET) using the Hargreaves equation (Hargreaves and Samani, 1982).

## 2.3 Carbon and water flux measurements

Carbon and water fluxes were measured at the two study sites from the 1st of January 2014 to the 31st of December 2022, using closed-path eddy covariance systems (CPEC200; Campbell Scientific Inc., Logan, Utah, USA; hereafter CSI). The eddy covariance systems were located in the southeast corner of the forest and in the center of the peatland (Fig. 1c).

All fluxes were computed using the EddyPro software, which allowed us to apply statistical, instrumental, footprint, and spectral corrections to the data. Secondly, we applied a post-processing methodology that included a quality screening of physically possible values, a first biometeorological gap-filling using linear regressions with ERA5 data as predictors, friction velocity threshold detection and filtering, and a general gap-filling approach (Marginal Distribution sampling (MDS), as described in Reichstein et al., 2005). We found that data with longer gaps (> 30 days) filled using MDS had a significantly lower $R^2$. Therefore, following a similar gap-filling technique as (Zhu et al., 2022), where they proposed a Machine Learning Random Forest method to improve gap-filling on longer gaps, we used a Machine Learning algorithm called Bagging Regression, using as a base estimator Random Forest (Bréda et al., 2006; Pedregosa et al., 2011).

The Bagging Regression model was able to reproduce unseen flux data with a $\sim 0.9$ score. Nevertheless, every year when this method of gap-filling was used is indicated in Table 3.

### 2.3.1 Energy Balance Closure and LE underestimations

To estimate the Energy balance closure, we calculated the Energy Balance Ratio (EBR) as:

$$EBR = \frac{\sum(LE + H)}{\sum(R_n - G)} \qquad (1)$$

The EBR was 0.7 and 0.93 for the forest and peatland ecosystems, respectively. The missing energy in both the peatland and forest ecosystems could be connected to the energy spent by photosynthesis, and the energy stored in the canopy structure, the soil water, and the litter. The lack of closure could also be understood as a systematic underestimation of H and LE related to synoptic-scale transport phenomena that are not captured by the EC systems (Mauder et al., 2013). There is a larger discrep-

135 ancy in energy balance closure in the forest as the magnitude of synoptic scale transport is likely to be larger in taller canopies and because the forest's canopy structure can store more energy due to a much larger biomass than the other ecosystem measured. Although this issue in the forest could be approached by having a longer averaging period, e.g. 60 minutes instead of 30 minutes, increasing the time average leads to other problems related to non-stationarity (Mauder and Foken, 2006).

Additionally, high relative humidity can produce an underestimation of LE, especially with closed-path systems, as the cut-off frequency of the closed-path system for water vapor concentration measurements decreases exponentially with increasing relative humidity (Zhang et al., 2023a).

Based on these assumptions, we implemented two corrections separately:

1. the Mauder et al. (2013) correction, hereafter the Bowen Ratio Correction (BRC), which uses the energy balance residual,
evaluated on a daily basis, to partition the residual between H and LE in a way that preserves the Bowen ratio.

2. the Zhang et al. (2023a) correction, hereafter High Relative Humidity Correction (HRHC), which rectifies LE considering the impact of high relative humidity.

Prior to the application of HRHC and BRC, a substantial 30% energy deficit was observed in the forest site. After the application of the HRHC, LE increased by only 6.1% in the peatland and 2.5% in the forest, while the calculated ET using HRHC

was smaller in the forest than in the peatland, contrary to expectations based on their canopy leaf areas. Due to observed limitations in the ability of HRHC to accurately capture LE variations in cases of poor EBR, we opted for the BRC over the HRHC.

Despite the acknowledged risk of potentially overestimating evapotranspiration using EBR, it provided a more robust correction approach compared to HRHC when comparing ET in both ecosystems. Furthermore, the decision to exclusively apply

the Bowen Ratio Correction (BRC) was influenced by the challenge of simultaneously using both corrections, as they operate on different principles and may introduce complexities in interpreting the corrected results. Nonetheless, we report the estimation of ET using both corrections (Supplementary material, Tables S2) and their partitioning values (Supplementary material, Tables S3 and S4).

## 2.4  Carbon and water flux partitioning

At the peatland site, where the canopy height is low and the meteorological conditions are appropriate for assuming a well-mixed boundary layer, the net ecosystem exchange of $CO_2$ (NEE) was assumed to be equivalent to the flux measured. For the forest ecosystem, where the canopy height is much higher and the conditions might result in heterogeneous boundary layer conditions when there is reduced turbulence, the NEE was calculated as the sum of the $CO_2$ flux and the storage term. The storage term was estimated from a single-point measurement using the EddyPro software. The NEE was partitioned into gross primary productivity (GPP) and ecosystem respiration (Reco) (expressed in g C m$^{-2}$ s$^{-1}$) using the 'nighttime method' proposed by Reichstein et al. (2005).

Evapotranspiration was measured by the eddy covariance technique and expressed in kg $H_2O$ m$^{-2}$ s$^{-1}$. The quality control and data screening required for the partitioning of ET into evaporation (E) and transpiration (T) followed a similar procedure to that described in Zhou et al. (2016), considering only half-hour data that met the following conditions:

1. daytime data, with PAR $> 5$ $\mu$ mol m$^{-2}$ s$^{-1}$ or incoming shortwave radiation $> 10$ W m$^{-2}$.

2. good quality half-hourly data, i.e., quality 0 and 1 according to Mauder and Foken (2004).

3. days with no precipitation.

4. data during the growing season were selected from each site to estimate the highest T/ET (uWUEp); the growing season was filtered into days when the average half-hourly GPP was at least 10% of the 95th percentile of all the half-hourly GPP for the site.

Daily values of carbon and water fluxes were calculated as the accumulation of the half-hourly available data, including only the days when there was ≥70% valid data.

The ET partitioning methodology is based on the concept of the underlying water use efficiency (uWUE) and uses the apparent and potential water use efficiency (uWUEa, uWUEp, respectively) to calculate the ratio of T/ET, as follows:

$$uWUEa = \frac{GPP}{ET}\sqrt{VPD} \tag{2}$$

$$uWUEp = \frac{GPP}{T}\sqrt{VPD} \tag{3}$$

where VPD is the vapor pressure deficit (measured in hPa), derived from Ta and RH measurements using the equations proposed by Monteith and Unsworth (1990) and Murray (1967). Hence,

$$T = \left(\frac{uWUEa}{uWUEp}\right)ET \tag{4}$$

The procedure assumes that uWUEp is constant for each flux site and uWUEa reaches its maximum value (uWUEp) when T is equal to ET for terrestrial ecosystems with a high vegetation cover during the growing season. Thus, both uWUEa and uWUEp,

and hence T and E can be estimated from half-hourly GPP, ET, and VPD measurements (Zhou et al., 2016). Specifically, uWUEa is calculated directly from Equation 2 using the available half-hourly data. As uWUEp represents the upper bound of the uWUEa, it is calculated using quantile regression for the 95th percentile.

As in Zhou et al. (2016), uWUEp is here assumed to be constant for each site and representative of it when it is calculated as a long-term average uWUEp. This can be assumed because the standard deviation of the annual values of uWUEp are only departed 8.5% and 3.3% from the long-term average of uWUEp in the forest and peatland, respectively.

## 2.5 Ecosystem water use efficiency

Ecosystem water use efficiency (WUE, $g\,C\,kg\,H_2O^{-1}$) was calculated as:

$$WUE = \frac{GPP}{ET} \tag{5}$$

An alternative parameter called the inherent water use efficiency (IWUE) was proposed by Beer et al. (2009) to account for the direct effect of VPD on surface conductance and defined as:

$$IWUE = \frac{GPP}{ET}VPD \tag{6}$$

A third way to calculate ecosystem WUE is called the intrinsic water use efficiency (iWUE, $\mu mol\,CO_2\,mol^{-1}H_2O$) (Lloyd et al., 2002) :

$$iWUE = \frac{GPP}{G_s} \tag{7}$$

where $G_s$ is the bulk surface conductance of the ecosystem ($mol\,m^{-2}\,s^{-1}$), calculated by inverting the Penman-Monteith equation using meteorological data (Lloyd et al., 2002). Finally, a fourth option to calculate the ecosystem WUE is the underlying water use efficiency (uWUEa, hereafter uWUE), described by Equation 2. As uWUE also includes VPD in the calculation, it also accounts for the effect of VPD on surface conductance.

## 2.6 Relationships among ET, E, T, GPP and water use efficiency

We calculated a Pearson Correlation Matrix to assess the relationship between water use efficiency, ET and its components, and GPP, using daily data that met the conditions described in section 2.4. Before calculating the correlations, the time series were detrended and their annual cycles were removed, thus the measured correlations only account for the anomalies and their direct impact on the other variables rather than including their annual patterns determined by environmental factors and their trends driven by global changes. The detrending and annual cycle removal were performed following the STL methodology described by Cleveland et al. (1990), using a periodicity of 365 days. The STL methodology is a seasonal-trend decomposition procedure based on a local polynomial regression which calculates the residuals of a time series after removing the seasonal trends.

## 2.7 Assessing the environmental drivers of the fluxes

We used structural equation modeling (SEM) to estimate the influence of Rg, Ta, VPD, U, P, SWC, and WTD on the dependent variables GPP, ET, E, and T. Analyses were conducted independently for the forest and peatland sites. We used a partial least squares path modeling (PLS-PM or PLS-SEM; Tenenhaus et al., 2005), a non-parametric composite-based SEM that has shown potential in analyzing large sets of ecological and environmental data (Ferner et al., 2018; Lopatin et al., 2015, 2019, 2022; Lopatin, 2023). PLS-PM uses ordinary least square regressions for estimating the path coefficients and has been found to be flexible to model interactions using a reflective or a formative conceptualization, which dramatically alters the way of the measurement approximation (e.g., effect indicator, causal indicator, or composite indicator). Through simulations, Sarstedt et al. (2016) found that PLS-PM and factor-based SEM do not differ significantly when reflective modes are used (the default for the PLS-PM and the only option for SEM). They also found that PLS entails practically no bias when estimating data from a composite model population, regardless of whether the measurement models are reflective or formative. Hence, PLS-PM is often robust if one is unsure of the nature of the data. We standardized the variables to normalize the path coefficients and intercepts (i.e., turn variables with different raw units into standard deviation units; Grace and Bollen, 2005). Stratified bootstrapping with 1,000 repetitions was used to ensure that all 30-minute data were equally drawn at every iteration and to assess significant interactions ($\alpha = 0.05$). For each iteration, the observations were randomly selected with replacements from the available samples per strata, from which 36.8% on average were not selected. We used these observations as holdout samples for validation (Kohavi, 1995). Bootstrapping helps to estimate the accuracy of the estimated coefficients and also their stability in the face of possible collinearity (e.g., Basagaña and Barrera-Gómez, 2022). Model performances were measured in terms of the coefficients of determination (r2; calculated as the squared Pearson's correlation coefficient) and the normalized root mean square error (nRMSE; expressed in percentage). The normalized root mean square error was calculated as:

$$nRMSE = \left( \frac{\sqrt{\frac{1}{n}\sum_{i=1}^{n}(y_j - \widehat{y_j})^2}}{max(X) - min(X)} \right) \times 100 \tag{8}$$

where X is the dependent variable. Finally, we used a one-sided bootstrap pair test to check for significant differences ($\alpha = 0.01$) between the forest and peatland models in terms of their accuracy and path coefficients (Lopatin et al., 2019). We used the R-package *plspm* for the SEM analyses.

## 3 Results

### 3.1 Seasonal patterns of environmental drivers

Solar radiation, air temperature and precipitation showed typical seasonal variations associated with this region (Fig. 2a). Lower temperatures and higher rainfall were associated with the austral winter (June-August) and higher temperatures and a lower rainfall with the drier and warmer period during the summer months (December-February) (Fig. 2b-c). Annual precipitation ranged between 1371 mm and 2490 mm, with the driest year (2016) coinciding with the highest observed mean global yearly

radiation and air temperature (Fig. 2). Based on the data collected since 1999 at the Senda Darwin meteorological station, we found no trend in the precipitation data but an increasing trend for potential evapotranspiration, although at a small rate (3.1 mm per year, Supplementary material Fig. S1).

The monthly values of the measured micrometeorological variables, shown separately for each ecosystem, are shown in Fig. 3. Annual values for these variables are included in the Supplementary material (Table S1). The net radiation was very similar
at both sites during the winter but higher in the forest during the summer (Fig. 3a) which could be explained by a higher albedo for the peatland in summer, when the *Sphagnum* change its color from green to yellowish. Wind speed at both sites was slightly higher in winter, but the internal variability was high enough to produce significant annual fluctuations that did not allow us to identify a clear seasonal behavior (Fig. 3b). Higher mean wind speeds were associated with the forest compared to the peatland, which is explained by the higher measurement height. Surface conductance was higher in winter, which could
be driven by the more turbulent conditions created by the higher wind speeds (Fig. 3c). Broadly, the VPD at both sites showed marked seasonal behavior with the lowest values during winter when the atmosphere reaches its maximum water saturation in this area. Variations in water table depth and soil water content indicate that drier soil conditions typically occurred during late summer and early fall. Higher soil moisture contents were observed between winter and early spring (Fig. 3d). Whilst the water table depth was closer to the surface in the peatland than in the forest, this was not reflected in a higher soil moisture
content in the peatland. Soil temperatures were similar in both ecosystems during the winter, but higher values were found in the peatland during summer (Fig. 3e).

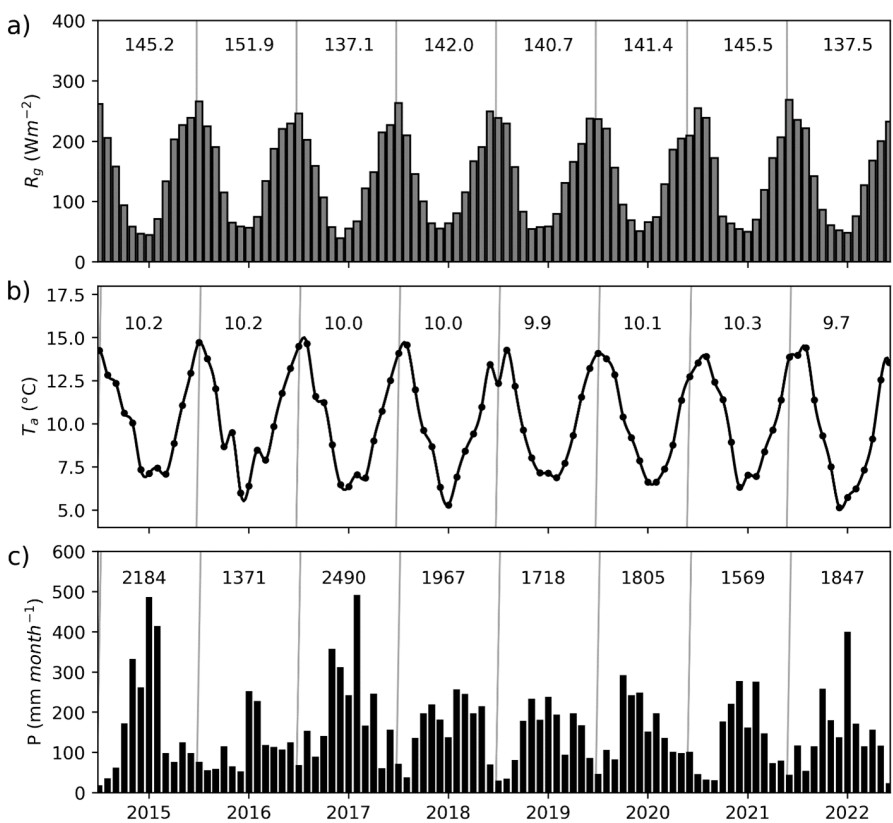

**Figure 2.** Monthly values of a) mean global radiation (Rg), b) mean air temperature (Ta), and c) cumulative precipitation (P) during the study period, as recorded at the Senda Darwin meteorological station. The numbers inside the panels represent the mean values for Rg, Ta and the annual sum for P for each year.

### 3.2 Seasonal and annual variation of carbon and water fluxes and water use efficiency

The daily means and annual variability of gross primary productivity (GPP), evapotranspiration (ET) and the different estimates of water use efficiency (WUE) are shown in Fig. 4, while the annual values are shown in Table 3. Both GPP and ET were highest
in the forest, with the greatest difference between the two ecosystems during the warmer months (Fig. 4a-b). This resulted in a 60% higher mean annual GPP in the forest ($1374 \pm 30$ g C m$^{-2}$ y$^{-1}$) compared to the peatland ($831 \pm 33$ g C m$^{-2}$ y$^{-1}$) but only a 33% higher annual ET ($910 \pm 59$ mm vs $682 \pm 25$ mm, Table 3). A consequence of the higher GPP without a corresponding increase in ET of a similar magnitude was that WUE was highest in the forest compared to the peatland, although there was no evidence of a seasonal pattern. The different forms of water use efficiency showed higher values for
the forest, with a mean value of WUE of $2.61 \pm 0.17$ (g C kg$^{-1}$H$_2$O) in the forest and $1.28 \pm 0.08$ (g C kg$^{-1}$ H$_2$O) in the peatland (Table 3).

In contrast to WUE, the inherent WUE (IWUE), intrinsic WUE (iWUE) and underlying WUE (uWUE) showed a seasonal

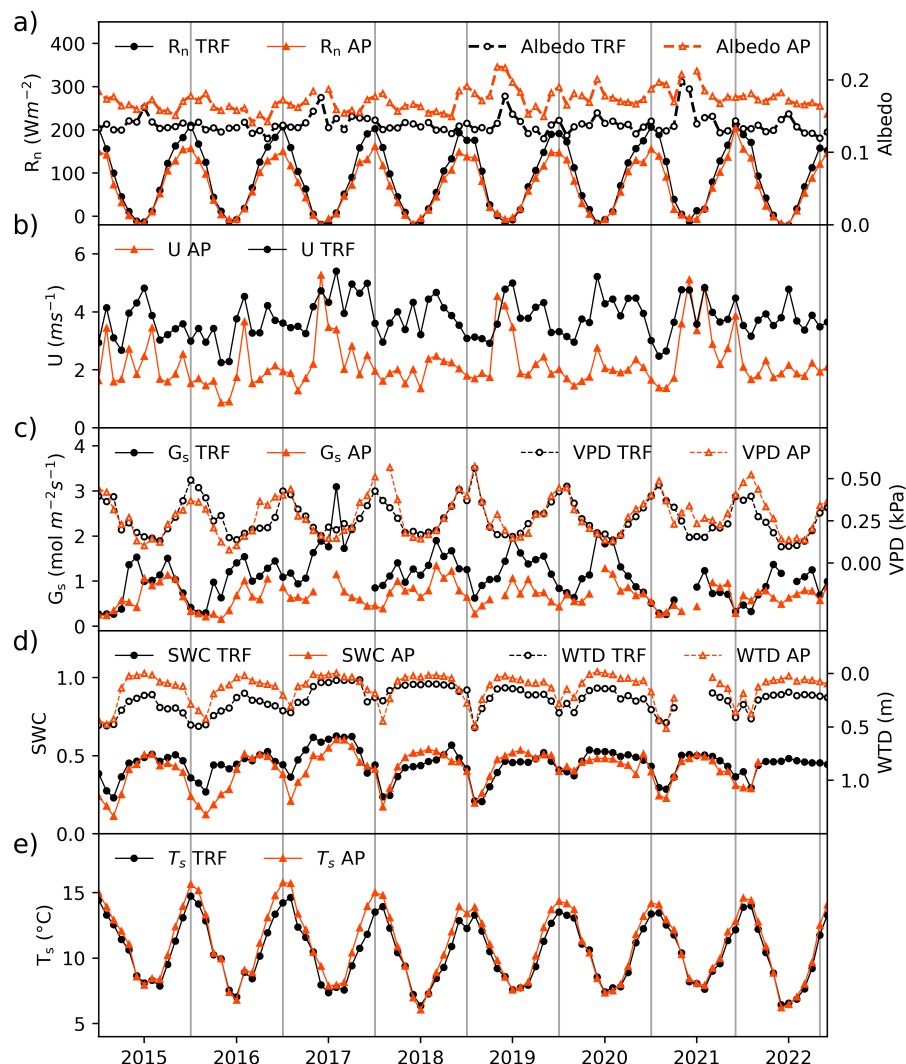

**Figure 3.** Monthly mean values of environmental variables in the temperate rainforest (TRF, black lines and symbols) and anthropogenic peatland (AP, orange lines and symbols) during the study period. Panels show a) cumulative solar radiation (Rg) and mean albedo, b) mean wind speed (U), c) surface conductance (GS) and vapor pressure deficit (VPD), d) mean soil water content (SWC) at 5 cm depth and water table depth (WTD), and e) mean soil temperature (Ts) at 5 cm depth.

pattern, with higher values during the warmer months (Fig. 4d). For all formulations the forest showed higher values than the peatland, although the difference was lower for iWUE and uWUE (Fig. 4e-f).

Based on the annual net ecosystem exchange (NEE) the forest was a consistent carbon sink ($-486 \pm 23$ g C m$^{-2}$ y$^{-1}$) while the peatland was, on average, a small source ($33 \pm 21$ g C m$^{-2}$ y$^{-1}$) (Table 3), which was the result of a much larger GPP for the forest compared to the peatland as the two ecosystems had similar ecosystem respiration rates, despite the higher biomass

of the forest. Possible explanations for this are that the higher temperatures in the peatland compensated for a lower labile C availability or because some of the C respired from the forest soil is assimilated by the canopy, therefore decreasing the ecosystem respiration rate. While the forest acted as a net carbon source for only two months in the autumn (May and June), the peatland was only a net sink during three spring months (September-November) (Supplementary material, Fig. S2). Transpiration represented, on average, 46% of ET in the forest and 48.6% in the peatland, although a significantly higher ET was found in the forest (Table 3). The monthly values of the contribution of T to ET in both ecosystems showed similar values and seasonality (Supplementary material, Fig. S3). The annual ET in the forest and peatland represented 43% and 32% of the mean annual precipitation, respectively.

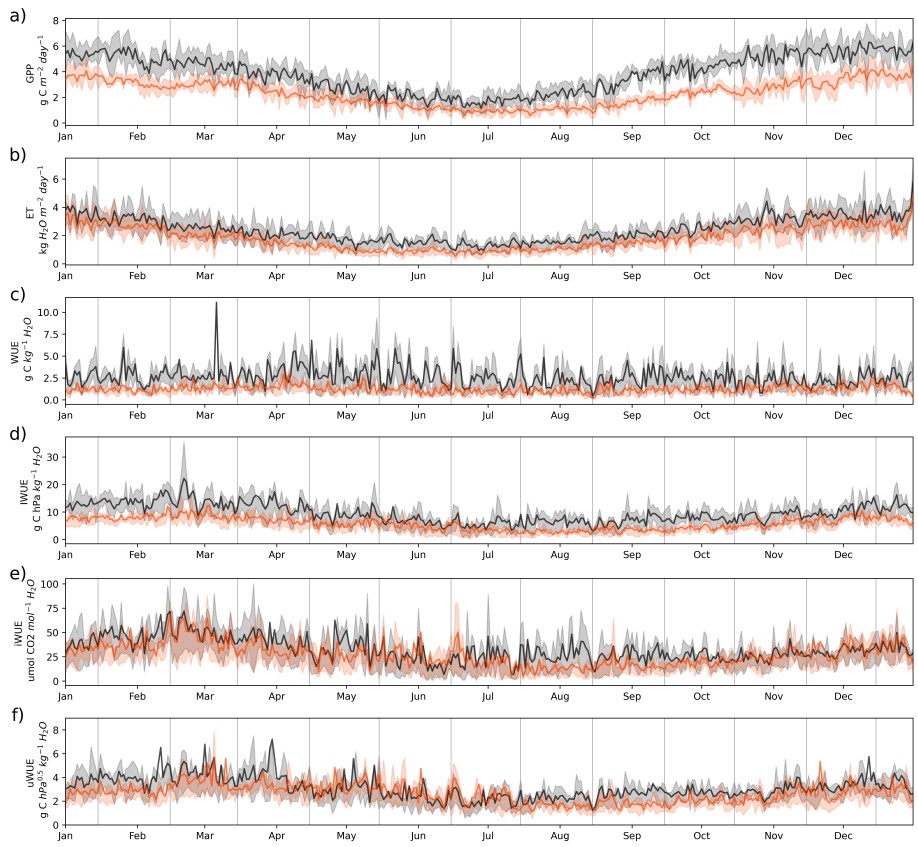

**Figure 4.** Daily values of (a) gross primary productivity (GPP), (b) evapotranspiration (ET), (c) water use efficiency (WUE), (d) inherent water use efficiency (IWUE), (e) intrinsic water use efficiency (iWUE) and f) underlying water use efficiency (uWUE) in the temperate rainforest (black lines) and in the anthropogenic peatland (orange lines) during the study period. The shading associated with the black and orange lines represents the variability between the 0.25 to 0.75 quantiles.


### 3.3 Relationship between carbon and water fluxes and WUE

The correlation matrices for carbon and water fluxes and WUE are shown in Fig 5. A high correlation was found between ET and evaporation, while transpiration was highly correlated with GPP; all the correlations were statistically significant and with a p-value<0.05. All forms of WUE showed that variations in ecosystem WUE were more correlated (negatively) to changes in ET ($r \leq -0.60$) than to changes in GPP (r between -0.16 and 0.24) and related more to E rather than T, with both ET and E negatively correlated to the different expressions of WUE. Although all forms of WUE were positively correlated with each other, the highest correlation was observed between IWUE and uWUE (r=0.94).

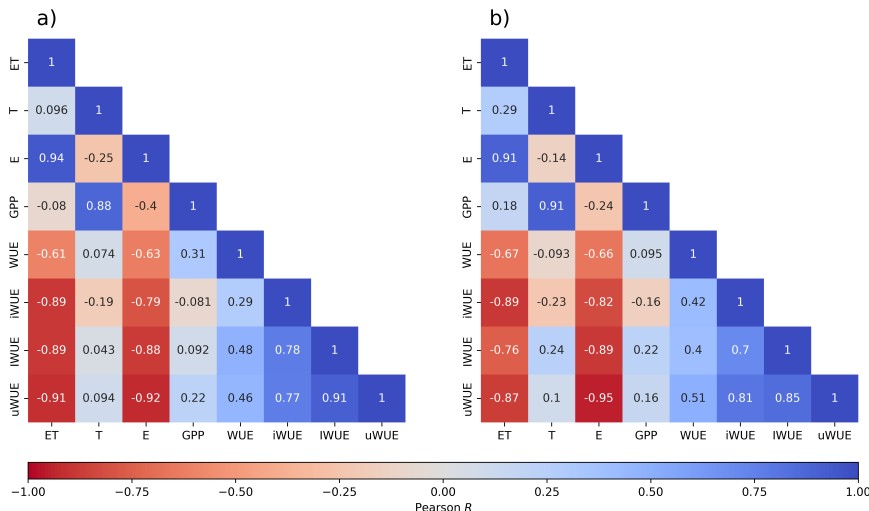

**Figure 5.** Pearson correlation matrix heatmap between water use efficiency and carbon and water fluxes for a) temperate rainforest and b) anthropogenic peatland.

### 3.4 Partitioning of evapotranspiration

Daily maximum rates of evapotranspiration reached 9 mm $\mathrm{day}^{-1}$ in the forest and 6.7 mm $\mathrm{day}^{-1}$ for the peatland (Fig. 6). The weekly contribution of T to ET varied widely around the mean values of 46% for the forest and 48.6% for the peatland, with the highest values of 89% for the forest and 84% in the peatland. The highest values of T/ET were observed in the summer months while the lowest values were found in the winter (Supplementary material Fig. S3).

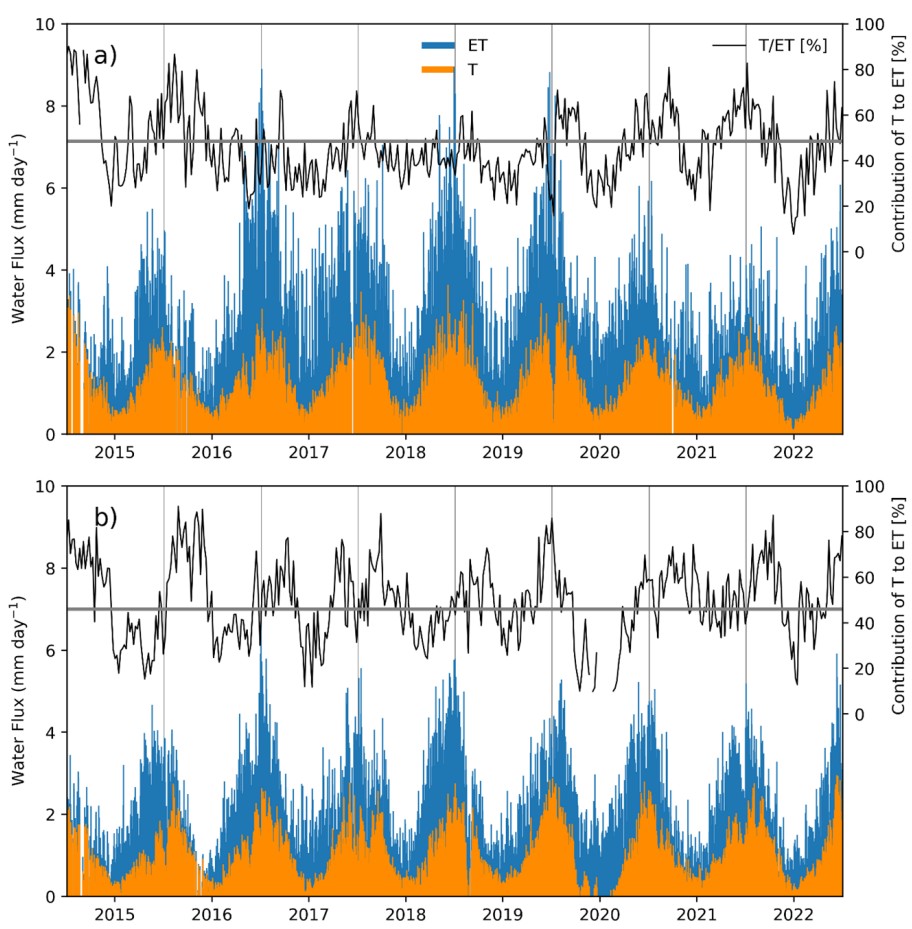

**Figure 6.** Daily measured evapotranspiration rates (ET) and estimates of transpiration (T) and their weekly contribution to ET for a) a temperate rainforest and b) an anthropogenic peatland, during the study period, based on the uWUE method (Zhou et al., 2016). The upper parts of each panel show variations in the mean contribution of T to ET in each ecosystem, with the average indicated by the horizontal line.

**Table 1.** Annual mean carbon and water fluxes and WUE in the temperate rainforest and anthropogenic peatland sites for the study period.

| Year | NEE (g C m⁻² y⁻¹) | Reco (g C m⁻² y⁻¹) | GPP (g C m⁻² y⁻¹) | ET (kg H₂O m⁻² y⁻¹) | E (kg H₂O m⁻² y⁻¹) | T (kg H₂O m⁻² y⁻¹) | T/ET (%) | Gs (mol m⁻² s⁻¹) | WUE (g C kg⁻¹ H₂O) | IWUE (g C hPa kg⁻¹ H₂O) | iWUE (μmol CO₂ mol⁻¹ H₂O) | uWUE (g C hPa⁰·⁵ kg⁻¹ H₂O) |
|---|---|---|---|---|---|---|---|---|---|---|---|---|
| | | | | | | | Temperate rainforest | | | | | |
| 2015* | -504 | 980 | 1484 | 745 | 320 | 306 | 55.8 | 0.82 | 3.09 | 10.8 | 42.9 | 3.9 |
| 2016 | -391 | 982 | 1373 | 820 | 444 | 305 | 49.0 | 0.91 | 3.29 | 11.2 | 37.1 | 3.5 |
| 2017 | -482 | 933 | 1415 | 1122 | 656 | 394 | 40.1 | 1.58 | 1.94 | 7.9 | 25.3 | 2.6 |
| 2018 | -538 | 933 | 1471 | 1089 | 611 | 437 | 42.6 | 1.28 | 2.35 | 8.3 | 29.3 | 2.5 |
| 2019 | -518 | 840 | 1359 | 1071 | 630 | 409 | 39.7 | 1.25 | 2.46 | 8.3 | 28.6 | 2.3 |
| 2020 | -585 | 789 | 1375 | 936 | 484 | 385 | 45.2 | 1.10 | 2.08 | 8.3 | 29.8 | 2.6 |
| 2021* | -410 | 853 | 1263 | 721 | 335 | 326 | 51.3 | 0.61 | 2.89 | 10.7 | 41.1 | 3.5 |
| 2022 | -462 | 794 | 1256 | 772 | 400 | 321 | 44.3 | 0.89 | 2.81 | 8.5 | 34.2 | 2.7 |
| X̄ ± SE | -486 ± 23 | 888 ± 29 | 1374 ± 30 | 910 ± 59 | 485±47 | 360±18 | 46.0±2.0 | 1.05 ± 0.11 | 2.61 ± 0.17 | 9.3 ± 0.5 | 33.5 ± 2.2 | 2.9 ± 0.21 |
| | | | | | | | Anthropogenic peatland | | | | | |
| 2015 | 112 | 885 | 773 | 584 | 313 | 204 | 48.2 | 0.65 | 1.37 | 5.7 | 34.2 | 2.8 |
| 2016 | 1 | 829 | 828 | 598 | 295 | 258 | 52.8 | 0.47 | 1.67 | 6.7 | 35.2 | 2.8 |
| 2017 | 8 | 911 | 903 | 738 | 368 | 326 | 47.5 | 0.72 | 1.14 | 5.0 | 24.7 | 2.3 |
| 2018 | 42 | 841 | 800 | 763 | 417 | 317 | 46.7 | 0.83 | 1.10 | 5.0 | 22.5 | 2.2 |
| 2019 | -19 | 830 | 849 | 651 | 307 | 307 | 51.2 | 0.64 | 1.09 | 5.4 | 27.1 | 2.5 |
| 2020* | -45 | 683 | 729 | 770 | 348 | 271 | 37.3 | 0.73 | 0.96 | 4.2 | 22.5 | 2.0 |
| 2021 | 114 | 951 | 837 | 670 | 331 | 311 | 52.6 | 0.52 | 1.12 | 6.2 | 29.0 | 2.5 |
| 2022 | 55 | 986 | 932 | 681 | 301 | 342 | 52.5 | 0.70 | 1.21 | 5.9 | 22.1 | 2.5 |
| X̄ ± SE | 33 ± 21 | 864 ± 33 | 831 ± 23 | 682±25 | 335±15 | 292±16 | 48.6±1.8 | 0.66 ± 0.04 | 1.28 ± 0.08 | 5.5 ± 0.3 | 27.2 ± 1.9 | 2.5 ± 0.10 |

Values of NEE, Reco, GPP, and ET are the yearly aggregation of 30-min frequency data, while the other variables are based on the yearly average. WUE, iWUE, and IWUE were estimated using half-hourly GPP, ET, VPD, or Gs data. Values of uWUE were calculated using daily average data based on the procedure used by Zhou et al. (2016). * Represents years where there was a significant amount of missing data, and although gap-filled, may have greater uncertainties. Transpiration (T) + Evaporation (E) may not be equal to ET every year because the partitioning model does not meet the requirements to calculate the partitioning every day.

### 3.5 Drivers of carbon and water fluxes

The structural equation models (SEM) showed that global radiation (Rg) had a significant ($\alpha = 0.01$) positive correlation with forest and peatland GPP (Fig. 7). In contrast, VPD, WTD and Gs had a negative influence on forest GPP, whilst Ta had a negative effect on peatland GPP. For both the forest and peatland ecosystems, Rg, VPD, and Gs had a positive effect on ET and a negative influence on ET. Assessment of the potential environmental drivers of E and T for both ecosystems indicated a significant positive effect of Rg and Gs on E, and a negative effect of WTD (Fig. 8). A positive impact of U on E was also found in the forest, while Rg and VPD positively affected T in the forest and peatland ecosystems.

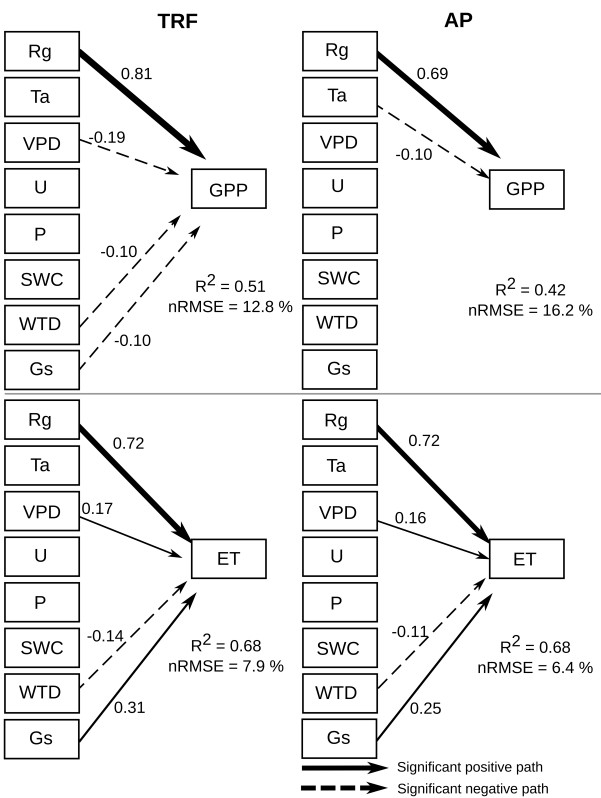

**Figure 7.** Structural equation model showing the influence of various environmental drivers on gross primary productivity (GPP) and evapotranspiration (ET) during the period 2014-2021 for an old-growth temperate rainforest (TRF) and an anthropogenic peatland (AP), using 30-minute data. Arrows represent significant ($\alpha = 0.01$) unidirectional relationships among variables. Solid and dashed arrows denote positive and negative relationships, respectively. The thickness of the arrows is scaled to reflect the magnitude of the path coefficient ($\beta$). All values correspond to the median value of the bootstrapping validation. The environmental drivers are solar radiation (Rg), air temperature (Ta), vapor pressure deficit (VPD), wind speed (U), precipitation (P), soil water content (SWC), water table depth (WTD), and surface conductance (GS).

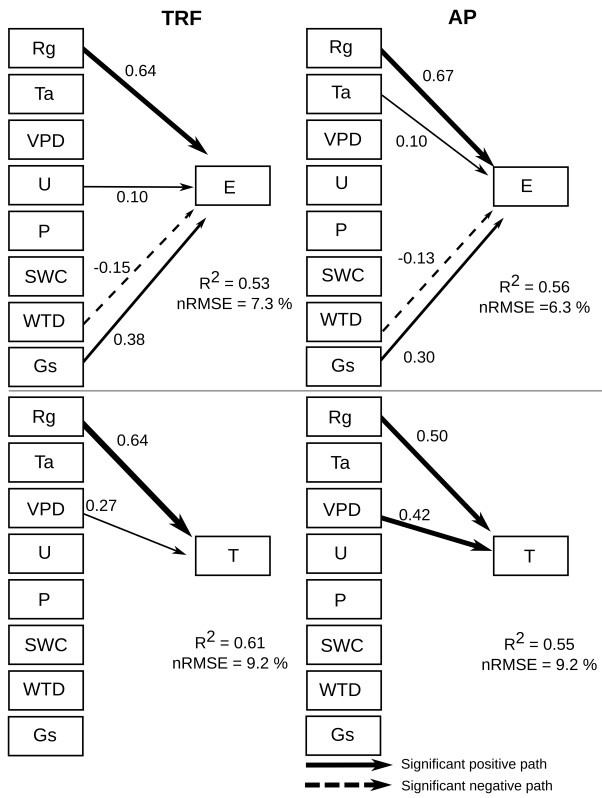

**Figure 8.** Structural equation model showing the influence of various environmental drivers on evaporation (E) and transpiration (T) during the period 2014 - 2021, using 30-minute data. Arrows represent significant ($\alpha$ = 0.01) unidirectional relationships among variables. Solid and dashed arrows denote positive and negative relationships, respectively. The thickness of the arrows is scaled to reflect the magnitude of the path coefficient ($\beta$). All values correspond to the median value of the bootstrapping validation. The environmental drivers are solar radiation (Rg), air temperature (Ta), vapor pressure deficit (VPD), wind speed (U), precipitation (P), soil water content (SWC), water table depth (WTD), and surface conductance (GS).

## 4 Discussion

### 4.1 Seasonal and annual variation of the fluxes and water use efficiency

The forest had higher water use efficiency values than the anthropogenic peatland based on all four ways of expressing WUE. The mean value of WUE for the forest ($2.61 \pm 0.17$ g C kg$^{-1}$H$_2$O) is similar to the mean value reported in the review by Zhang et al. (2023b) for 13 evergreen broadleaved forest sites ($\sim 2.5 \pm 0.5$ g C kg$^{-1}$H$_2$O) but slightly lower than the values reported by Beer et al. (2009) for evergreen broadleaved forests in Europe at similar latitudes (site ID: FR-PUE and IT-CPZ) ($\sim 3.3$ g C kg$^{-1}$H$_2$O). We found more significant differences for IWUE ($9.3 \pm 0.5$ g C hPa kg$^{-1}$H$_2$O) compared to northern hemisphere evergreen broadleaved forests ($32.02$ g C hPa kg$^{-1}$H$_2$O; Liu et al., 2022) including European sites ($30.61$ g C hPa kg$^{-1}$H$_2$O (Beer et al., 2009). Such differences are likely related to the negative relationship between IWUE and annual precipitation reported for this type of forest (Liu et al., 2022). The northern sites are located in much drier areas than ours (780 and 883 mm against 2100 mm), and this highlights the necessity of monitoring these ecosystems under a wider range of environmental conditions globally before any general conclusions can be drawn.

Similarly, our estimation of WUE at the peatland site ($1.28 \pm 0.08$ g C kg$^{-1}$H$_2$O) is at the lower end of the range of values reported for wetlands in Europe (1.23 and 1.73 g C kg$^{-1}$H$_2$O; Beer et al., 2009) and a peatland site in Oregon ($\sim 1.8$ g C kg$^{-1}$H$_2$O, Brümmer et al., 2012), likely because these sites have a much lower precipitation (between 395 and 894 mm). In turn, our estimation of IWUE ($5.5 \pm 0.3$ g C kg$^{-1}$H$_2$O) is comparable to a fen in Finland (site ID: FI-Kaa, 4.58 g C kg$^{-1}$H$_2$O; Aurela et al., 2004); even though the northern site has a lower precipitation (470 mm), with both sites showing similarities in vegetation, including the presence of mosses, sedges and shrubs.

Among the different formulations of ecosystem water use efficiency, WUE did not show a seasonal pattern. This supports the suggestion that expressions which account for changes in atmospheric water demand and minimize the influence of non-stomatal water fluxes, such as soil and canopy surface evaporation, should be used (Lavergne et al., 2019). Accordingly, IWUE, iWUE and uWUE showed higher values during the warmer and drier months of the southern summer (December-February), when the plants are more photosynthetically active and capture more carbon, making these expressions more suitable for representing water use efficiency at the ecosystem level.

In general, the values of WUE for our study sites were comparable to those from Northern Hemisphere sites, but when vapor pressure deficit was accounted for, our values were lower, which is likely explained by the higher precipitation/higher humidity under our conditions. Another reason may be that the observed effect of nitrogen deposition on increasing WUE, through its effect on GPP (Masri et al., 2019) may not play a role in our study site. The ecosystems of southern South America, where incoming weather fronts originate directly over the South Pacific Ocean, have not been exposed to industrial pollution, in contrast to the forests of the northern hemisphere, so the nitrogen cycle in the study area has been defined as unpolluted (Hedin et al., 1995). Annual values of WUE showed a wide variation in the forest (between 1.94 and 3.29 g C kg$^{-1}$H$_2$O) compared to the interannual variability reported for an old-growth subtropical forest over 7 years (between 1.70 and 1.98 g C kg$^{-1}$H$_2$O; Liu et al., 2017), which may be explained by the occurrence of an extremely dry year in our study site in 2016 (Garreaud, 2018), which was associated with the highest WUE value, followed by a wet year in 2017 that was associated with the lowest

WUE value.

Based on measurements of NEE the forest was a net sink, while the peatland was a small source, which is consistent with an estimation made by Perez-Quezada et al. (2021b), who showed that fire in this area has strong and long-term effects on the forest GHG balance, by converting the forest into an anthropogenic peatland. Our estimation of the average forest NEE based on eight years of measurement (-486 $\pm$ 23 g C m$^{-2}$year$^{-1}$) is higher than the estimation made for this forest during two previous seasons (-238 $\pm$ 31 g C m$^{-2}$year$^{-1}$; Perez-Quezada et al., 2018) and also higher than the value reported for a

coniferous rainforest in southern Chile from three years of measurements (-287 $\pm$ 38 g C m$^{-2}$year$^{-1}$; Perez-Quezada et al., 2023). Our estimation of NEE for the peatland showed that this ecosystem is a small carbon source (33 $\pm$ 21 g C m$^{-2}$year$^{-1}$), with a range between 114 and -45 g C m$^{-2}$year$^{-1}$, which contains the estimation (-22 g C m$^{-2}$year$^{-1}$) from a previous study at this site (Valdés-Barrera et al., 2019). We looked at the correlation between annual NEE and environmental variables but found no significant relationships (data not shown).

Both gross primary productivity (GPP) and evapotranspiration (ET) showed higher values in the forest compared to the peatland, with a much larger difference in GPP than ET, which could reflect greater stomatal control of water loss in the forest. In this case, ecosystem respiration did not differ significantly between the ecosystems (Table 1), which implies that the difference in NEE between ecosystems is explained by differences in GPP ($\sim$ 1374 g C m$^{-2}$year$^{-1}$ in the forest and $\sim$ 831 g C m$^{-2}$year$^{-1}$ in the peatland). Thus, the impact of the burning of the forest and resulting flooding that turned it into an

anthropogenic peatland is considerable, reducing the ecosystem $CO_2$ fixing capacity by $\sim$ 40%. Although the contribution of transpiration to ET is similar in both the forest and the peatland ($\sim$ 47%), annual ET is 35% larger in the forest than in the peatland. This finding could be partially explained because evaporation in the forest does not occur only from the soil but also from the canopy due to the canopy interception of the high precipitation in the area.

### 4.2 Evaporation as the main driver of evapotranspiration and water use efficiency

Evapotranspiration represented $\sim$ 40% of annual precipitation in the forest and $\sim$30% in the peatland, which are comparable to the value estimated for the forest in this area through modelling ($\sim$32%, Gutiérrez et al., 2014), but rather low compared to the 85% reported for a mixed boreal forested catchment in Sweden (Kozii et al., 2020). These relatively low values are likely related to the high precipitation in the study area and to the low mean daytime global radiation ($\sim$140 Wm$^{-2}$), although the mean precipitation for the study period (1868 mm) was lower than the historical mean ($\sim$2100 mm year$^{-1}$), which is consis-

tent with the drought observed in the last decade in central Chile (Garreaud et al., 2017). These environmental conditions may also explain the lower contribution of transpiration (compared to evaporation) to ET in both ecosystems ($\sim$47%), which is low compared to a temperate mixed forest in Belgium, where transpiration accounted for 58% of ET (Soubie et al., 2016).

A similar contribution of transpiration to ET in both ecosystems occurs despite quantitatively different flux values, with ET in the forest 33% higher on average than in the peatland. Although the leaf area index (LAI) has not been estimated for the

peatland site, we are certain that the LAI for the forest (3.7, Perez-Quezada et al., 2021) is much higher, representing a larger surface for evaporative and transpiration processes.

An important role for the interception and storage of precipitation by foliage, stems, epiphytic mosses, and lichens was previ-

ously reported as factors that would increase the evaporation component of ET in a tall old-growth forest in Oregon (Unsworth et al., 2004). A review on the hydrology of Chilean forests reported that interception ranged between 11 - 36% (Balocchi et al., 2023), with our study site value located at the higher end (33%, Frêne et al., 2022).

The large contribution that E makes to ET (Figure 6) may be explained by the frequently waterlogged and humid conditions in this area, where water can evaporate directly from soils and wet vegetated surfaces This, together with the presence of plant species that either do not have stomata (i.e., mosses in the peatland or epiphytes in the forest) or, in the forest, have poor stomatal control results in a decoupling of WUE from transpiration and GPP. A previous study showed that one of the dominant species in the forest (Drimys winteri) has traits that are focused more on efficient water transport favoring carbon gain over the ability to regulate water loss (Negret et al., 2013). In these wet high-latitude conditions, the evaporation process was more relevant, as it contributed a higher proportion of ET and was more correlated to ET compared to transpiration (Figure 5). This is consistent with the report from a northern peatland where evaporation accounted for about two thirds of the water flux when the surface of *Sphagnum* was wet (Kim and Verma, 1996). The generally greater contribution of evaporation to ET indicates that under humid, high rainfall conditions ecosystem water use efficiency (for all formulations) may be more related to evaporation than to transpiration. Although the use of different approaches to partition ET can sometimes yield different results (Nelson et al., 2020), a recent study (Melo et al., 2021) using remote sensing ET models estimated similar evaporation and transpiration fractions for our site. Nevertheless, independent estimations of transpiration using for example the sap flow method would help validate the ET partitioning modeling method used here.

We evaluated the partitioning method proposed by Nelson et al. (2018) to compare it with the method used in our work Zhou et al. (2016) and found that the former method yielded an even higher contribution of evaporation to ET. The results are shown in the Supplementary material, Tables S3 and S4. Furthermore, we found that the relationships between biometeorological variables and evaporation and transpiration fluxes were consistent between both methods (data not shown). While acknowledging the potential for variations and complexities when applying different partitioning methods in natural ecosystems, we think this supports our results.

### 4.3 Main environmental drivers of carbon and water fluxes

As expected for radiation-limited ecosystems, global radiation was the main driver for both GPP and ET, although the variables related to atmospheric water demand (vapor pressure deficit and surface conductance) were still important. These three variables were also identified as the main drivers of ecosystem carbon and water fluxes for northern peatlands in Canada (Humphreys et al., 2006). Liu et al. (2017) also found global radiation and vapor pressure deficit to be among the main drivers of GPP and ET in a subtropical forest in China, although they reported that precipitation and soil moisture were even more important.

In turn, global radiation was the main driver of evaporation and transpiration in both ecosystems. The SEM analysis showed that surface conductance was not significantly related to transpiration, suggesting a decoupling of transpiration from control by stomata, which was observed also in a highly humid tropical forest in Panama (precipitation 2600 mm; Meinzer et al.,

1995). This implies that transpiration at the site is influenced more by the humidity gradient (represented by VPD) between the ecosystem-atmosphere continuum, while evaporation is controlled primarily by the boundary layer conductance, which depends on wind speed and friction velocity. The decoupling of transpiration from stomatal conductance may also be related to the hydraulic conductance of the soil-root-leaf pathway (Meinzer et al., 1995), which can be increased by waterlogging (Kubota et al., 2023), a condition frequently observed in our study area. The positive effect of wind speed on evaporation in the forest (and not in the peatland) is likely related to the larger evaporating surface (leaf area index) and roughness in the tall canopy. Although a previous study showed that wind speed can be positively related to WUE at the leaf level (Schymanski and Or, 2016), the authors associated this effect to more efficient convective cooling under high solar radiation loads, which may not be as important in wet light-limited environments.

As the soils in this area are very shallow ($\sim$0.5 m), water table depth could also be a factor in determining GPP in the forest and ET in both ecosystems. This variable was negatively related to evaporation, meaning that a water table closer to the soil surface increased evaporation fluxes, which occurred during winter (humid) months ($\sim$0.2 m in the forest and $\sim$0.1 m in the peatland), when saturated soil conditions exposed more water to evaporation.

Even though no trend was found for annual precipitation in the study area since 1999, models predict a decrease in precipitation in the coming decades (Almazroui et al., 2021). Using a dynamic model with a projected increased drought, Gutiérrez et al. (2014) predicted a 15% decrease in ET and a reduction of 27% in aboveground biomass for our forest site. This means that if the observed increasing trend for potential ET continues, drier soil and atmospheric conditions are expected for more extended periods during the summer, which in turn could result in an increase in the contribution of transpiration to ET in the future. This scenario would also be associated with an increase in vapor pressure deficit, with a consequent increase in WUE in both ecosystems (Zhang et al., 2019), particularly in the forest, where stomatal control is a more significant factor. In the peatland, the observed positive effect of air temperature on evaporation could make the role of evaporation even more important. On-going efforts to mechanistically model the functioning of these ecosystems will help us to better predict the effects of climate change in this part of the world.

## 5    Conclusions

We found that both GPP and ET were higher in the forest compared to the peatland, although the difference was larger for GPP, suggesting a greater control of water loss in the forest. Among the four expressions of WUE that we calculated, only those that included atmospheric water demand showed seasonal variation, making their use more biologically relevant than estimates based on GPP/ET. The values found for WUE were low in the peatland and the forest compared to similar ecosystems in other parts of the world, which is likely explained by high annual precipitation/humidity at these sites in southern South America. This is also likely to be the explanation for why variations in ecosystem WUE were linked to changes in ET more than to changes in GPP and variations in ET and WUE were related more to changes in evaporation than to transpiration. As global radiation and surface conductance were the main drivers of evaporation, we expect that WUE may increase in the future in these ecosystems, particularly in the forest where stomatal control is likely to be more significant.

*Data availability.* The datasets are available at the AmeriFlux site with the code names CL-SDF for the forest and CL-SDP for the peatland.

*Author contributions.* Jorge Perez-Quezada: Conceptualization, Funding acquisition, Investigation, Methodology, Project administration, Writing - original draft preparation, Writing - review and editing; David Trejo: Data curation, Formal analysis, Investigation, Writing - original draft preparation; Javier Lopatin: Formal analysis, Writing - original draft preparation David Aguilera: Data curation, Investigation, Writing - original draft preparation; Bruce Osborne: Methodology, Writing - review and editing; Mauricio Galleguillos: Conceptualization,
Funding acquisition, Investigation, Methodology, Writing - review and editing Luca Zattera: Formal analysis, Writing - original draft preparation; Juan L. Celis-Diez: Conceptualization, Funding acquisition, Writing - review and editing; Juan J. Armesto: Conceptualization, Funding acquisition.

*Competing interests.* The authors declare that they have no conflict of interest.

*Acknowledgements.* This study was funded by the National Agency of Research and Development (ANID, Chile) through the grants
FONDECYT 1211652 (JPQ), FONDEQUIP AIC-37, PIA/BASAL FB210006 to the Institute of Ecology and Biodiversity (IEB), PIA/BASAL FB210018 to the Cape Horn International Institute (CHIC) and FONDAP CR2 15110009 Center for Climate Resilience Research. This research was partially supported by the supercomputing infrastructure of the National Laboratory for High Performance Computing Chile (NLHPC) (ECM-02). This is a contribution to the Research Program of Senda Darwin Biological Station and the Chilean Long-Term Socio-Ecological Research Network (LTSER-Chile), affiliated with ILTER, AmeriFlux, and FLUXNET.

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
