# Peer review of "Comparison of carbon and water fluxes and the drivers of ecosystem water use efficiency in a temperate rainforest and a peatland in southern South America"

_EGUsphere, 2023_

## Author Response (AR3)

**Reply to Referee #1**

The work is well structured, well written, and easy to read because it is well told. It presents data on water and carbon exchange, using the eddy covariance technique for 8 years, from a temperate forest in northern Patagonia in Chila and a neighboring artificial peatland. This is very valuable since data of this type is not abundant in the region.

Its scientific significance and quality are excellent. The presentation quality is good.

The methods used to respond to the stated objectives are correct. The results presented are clear and the conclusions clearly come from their analysis.

R: We appreciate the positive comments from the referee about our manuscript.

Although it is possible to calculate primary productivity and GPP and the authors do so, since it is one of the objectives of the work, the title of the work does not mention it. I do not know if the authors plan to publish another work where they carry out an in-depth analysis of the information generated, it would be a shame not to do that exercise.

R: Because both referees mentioned that the title does not include all the elements that we analyze, we propose to change the title to: "Comparison of carbon and water fluxes and the drivers of ecosystem water use efficiency in a temperate rainforest and a peatland in southern South America"
Given that we do discuss GPP, we assume that the reviewer refers to NEE when asking for an in-depth analysis. Therefore, to emphasize that we do analyze these fluxes, we included the average NEE of both ecosystems in the abstract using the following words, "the forest was a consistent C sink (-486 ± 23 g C m−2 y−1) while the peatland was, on average, a small source (33 ± 21 g C m−2 y−1)".  A more in-depth analysis would require further analyses and extending the manuscript.

In the same sense, there is a lack of discussion on the anthropic effect of modifying the native forest and its consequences in terms of water and carbon dynamics. I am not so interested in changes in water use efficiency (in various versions) in an ecosystem in which water is not a limiting factor.
 R: We address the topic of anthropogenic effect by indicating that fires, associated with anthropogenic activities in this area, could result in a marked change in land cover and associated ecosystem services, converting these forests to peatlands with a consequent reduction in the C sink and changes in hydrology and water fluxes.

In the Discussion section we mention that while the forest acts as a sink for $CO_2$ of about 486 g C m-2 y-1, the anthropogenic peatland behaves as a small source of carbon of 33 g C m-2 y-1. We added a sentence and a reference to emphasize that fires can have strong and long-lasting effects on the GHG balance of forests in this area (Lines 322-324): "Based on measurements of NEE the forest was a net sink, while the peatland was a small source,

which is consistent with an estimation made by Perez-Quezada et al. (2021b), who showed that the effect of fire on the forest has strong and long term effects for the GHG balance in this area, by converting the forest into an anthropogenic peatland"

We also added an analysis in the Discussion section about the differences between ecosystems for GPP, Reco, and ET (Lines 334-339): "In this case, ecosystem respiration did not differ significantly between ecosystems (Table 1), which implies that the difference in NEE between ecosystems is explained by GPP (~ 1374 g C m-2 y-1 in the forest and ~ 831g C m-2 y-1 in the peatland). Thus, the impact of the burning of the forest and resulting flooding that turned it into an anthropogenic peatland is considerable, reducing the ecosystem's $CO_2$ fixing capacity by ~40%. Although the contribution of transpiration to ET is similar in both the forest and the peatland (~47%), annual ET is 35% larger in the forest than in the peatland.

This suggests that higher LAI in the forest does not only relates to higher transpiration rates but also to higher evaporation rates from the canopy due to the canopy intercepting significant amounts of water, associated with the high precipitation in this area.

Regarding the use of different formulations of WUE, we showed that the different terms used for WUE helped to identify the effects of VPD. Even though soil water availability may not be limiting in this area, changes in VPD could independently effect GPP and WUE.

Perez-Quezada, J. F., Urrutia, P., Olivares-Rojas, J., Meijide, A., Sánchez-Cañete, E. P., & Gaxiola, A. Long term effects of fire on the soil greenhouse gas balance of an old-growth temperate rainforest. Science of the Total Environment, 755, 142442. https://doi.org/10.1016/j.scitotenv.2020.142442, 2021

There are some minor errors regarding the use of parentheses when citing the bibliography (such as on page 2, paragraph 35, when citing Teran et al 2023). I encountered this error several times.
R: All these errors have been corrected.

In Figure 1, it would be clearer that the limits of Chile were present (fig 1a) and an arrow should appear indicating North, as in Figs 1 b and 1c.
R: The borders of Chile and other countries were added, same as the north symbol to Figures 1 a, b, and c.

Equation 4 has an error, since instead of WUE P, they wrote WUE a in the numerator and denominator.
R: Thanks for catching this error. The correct equation is T=(uWUEa/uWUEp) ET.

**Reply to Referee #2**

This paper implemented comparison between one forest and its associated peatland site in terms of GPP, ET (and its component), and different expressions of water use efficiency, using 5-yr closed-path eddy covariance data. The work could provide some insights into the physiological perspective of these ecosystems as there are not as rich datasets as in other regions. However, there are a number of points that should be improved to clarify the implications of this comparison for our understanding in these two different ecosystems.

1. Title. Only one third of the work looking at the drivers, good to mention the comparison of WUEs and fluxes.

R: Because both referees mentioned that the title does not include all the elements that we analyze, we propose to change the title to: "Comparison of carbon and water fluxes and the drivers of ecosystem water use efficiency in a temperate rainforest and a peatland in southern South America"

2. There is definitely space to improve the logic of the introduction. From the current version, seems that this work is more a diagnostic on fluxes and WUEs at these wet sites, and the reasons of doing the ET partitioning and WUEs (point to physiological understanding) and different expressions of WUE (stomatal response to VPD) need be clearly stated.

R: We added text to the Introduction to state that we use different formulations of WUE to increase our understanding of the physiological processes and drivers involved in the exchange of carbon and water at the ecosystem level (Lines 43-46): "Besides the basic form of WUE (GPP/ET), other formulations include the effect of vapor pressure deficit (Beer et al., 2009) or the bulk surface conductance of the ecosystem (Lloyd et al., 2002), which can increase our understanding of the physiological processes involved in the exchange of carbon and water at the ecosystem level."

Similarly, in Lines 48-50 we explain how the partitioning of ET allows us to differentiate between biological (T) and physical (E) drivers of evaporative losses.

3. The diagnostic of EBR is totally missing! In such wet sites (using closed-path sensors), the energy balance closure problem could has large effect on the estimation of ET, ET partitioning and the estimations of WUE. the equation of EBR is described but not diabosed at all. Also need to check the relative humidity dependent water flux estimates as it influences not only the ET and T but also WUE (see Zhang et al. (2023)).
Zhang W., et al., The effect of relative humidity on eddy covariance latent heat flux measurements and its implication for partitioning into transpiration and evaporation Agric. For. Meteorol., 330 (2023), 109305, 10.1016/J.AGRFORMET.2022.109305

R: Thank you for your comment. We have now incorporated the EBR diagnostic and how we managed the data regarding the non-closure of the energy balance.

We also included an explanation of how we dealt with the relative humidity influence on latent heat fluxes. Our processing workflow already included a correction that addressed this issue, so we added a paragraph to explain this:

"The EBR was 0.7 and 0.93 for the forest and peatland ecosystems, respectively. The missing energy in both the peatland and forest ecosystems could be connected to the energy used by photosynthesis, and the energy stored in the canopy structure, the soil water, and the litter. The lack of closure could also be understood as a systematic underestimation of H and LE related to synoptic-scale transport phenomena that are not captured by the EC systems (Mauder et al, 2013). There is a larger discrepancy in energy balance closure with the forest, as the magnitude of synoptic scale transport is likely to be larger in taller canopies and as the forest's canopy structure can store more energy because of a much larger biomass than the other ecosystem measured. Although this issue in the forest could be approached by having a longer averaging period, e.g. 60 minutes instead of 30 minutes, increasing the time average leads to other problems related to non-stationarity (Mauder and Foken, 2006). Based on these assumptions, we implemented the Mauder et al, (2013) correction which uses the energy balance residues, evaluated daily, to partition that energy between H and LE in a way that preserves the Bowen ratio. High relative humidity could produce an underestimation of latent heat, especially with closed-path systems as the cut-off frequency of the closed-path system for water vapour concentration measurements decrease exponentially with increasing relative humidity (Zhang et al., 2023). Therefore, we implemented the correction suggested by Ibrom et al. (2007), which is available as a correction for low pass filtering effects in EddyPro."

4. Figure 4, which year of the data are? From the data and method section, the starting year is 2014, but here the time label is 2012. Please check. From the figure caption, seems that you plotted the multi-year mean across the data period of 2014-2022.
 R: Figure 4a had the wrong X-axis label; should be months, the same as the rest of the panels. This has been corrected.

5. Line 227-230, the clear seasonality in IWUE, iWUE, and uWUE is due to the inclusion of VPD, and the smaller difference in iWUE and uWUE is due to the lower weight of VPD. From here I was wondering the reason to compare these different expressions of WUE rather than the plant water use efficiency (GPP/T).
R: This is related to another question raised above. We think the reason to compare different formulations of WUE is explained by the sentence we added in the Introduction (Lines 43-46): "Besides the basic form of WUE (GPP/ET), other formulations include the effect of vapor pressure deficit (Beer et al., 2009) or the bulk surface conductance of the ecosystem (Lloyd et al., 2002), which can increase our understanding of the physiological processes involved in the exchange of carbon and water at the ecosystem level."

Regarding the suggestion about GPP/T, both GPP and T decrease in winter, although the second one decreases more. A significant decrease in T in winter could be explained by a

more humid environment, especially in this area where the atmospheric water content is regularly very close to saturation. Although the figure below shows that this formulation of WUE has a different seasonal pattern, we think it is consistent with the significant relation between VPD and T we described.

[Figure]

Orange - > Peatland; Black -> Forest

6. line 231-235, this is interesting, but why do the forest and peatland have similar respiration rate?

R: We think the similar rates of ecosystem respiration in the peatland and the forest, despite the higher root and aerial biomass of the forest, could be because the higher soil temperatures in the peatland compensated for a lower labile C availability. Also, assimilation of some of the C derived from soil respiration by the more extensive forest canopy could reduce ecosystem respiration.

We modified the text to (lines 259-262): "…the two ecosystems had similar ecosystem respiration rates, despite the higher biomass of the forest. Possible explanations for this are that the higher temperatures in the peatland compensated for a lower labile C availability or because some of the C respired from the forest soil is assimilated by the canopy, therefore decreasing the ecosystem respiration rate."

7. From the analysis, I was wondering how does the model account for the colinearity between variables? For example in Figure 5 and 6, I do expect to see a connection between precipitation with ET, T, or E, but obviously the model did not detect the connection.

R: Thank you for your insightful query regarding how our PLS path modeling (PLS-PM) accounts for collinearity among variables, specifically referring to Figures 7 and 8. Your observation about the potential link between variables like precipitation, ET, T, and E is indeed pertinent. In the PLS-PM model, a high collinearity among variables is addressed

through the construction of latent variables (LVs). These LVs, formed in a supervised manner akin to regression PLS or PCA, and effectively distill the shared variance of the highly collinear observed variables, thereby mitigating the impact of multicollinearity on the model's outputs. Unlike PCA, the LVs in PLS-PM are designed to maximize the explained variance in relation to the model's dependent constructs, offering a more directed approach to handling collinearity.

Regarding the single variables in our model, the issue of collinearity is further addressed during the bootstrapping process. To elaborate, our methodology involved the stratified resampling of our dataset 1,000 times, and recalculating path coefficients for each sample. This technique not only estimates the accuracy of these coefficients but also their stability in the face of collinearity. The bootstrap samples allowed us to construct confidence intervals for each path coefficient. These intervals did not encompass zero for our key relationships, further affirming their statistical significance despite potential collinearity.

We added this information in the Methods section as (Lines 213-214): "*Bootstrapping helps to estimate the accuracy of the estimated coefficients and also their stability in the face of possible collinearity (e.g., Basagaña and Barrera-Gómez, 2022).*"

Xavier Basagaña, Jose Barrera-Gómez, Reflection on modern methods: visualizing the effects of collinearity in distributed lag models, International Journal of Epidemiology, Volume 51, Issue 1, February 2022, Pages 334–344, https://doi.org/10.1093/ije/dyab179

Besides, the phenomenon that the significant impact of Gs on ET and E but no impact on T suggests that a decoupling between surface conductance and stomatal conductance, this could be true in such a wet sites, but I think these kind of aspects are really missing in the discussion part.
R: To analyse this topic more deeply, we rephrased those lines to (lines 379-386):

"In turn, global radiation was the main driver of evaporation and transpiration in both ecosystems. The SEM analysis showed that surface conductance was not significantly related to transpiration, suggesting a decoupling of transpiration from control by stomata, which was also observed in a highly humid tropical forest in Panama (precipitation 2600 mm; Meinzer et al., 1995). This implies that transpiration at the site is more influenced by the humidity gradient (represented by VPD) between the ecosystem-atmosphere continuum, while evaporation is controlled primarily by the boundary layer conductance, which depends on wind speed and friction velocity. The decoupling of transpiration from stomatal conductance may also be related to the greater influence of hydraulic conductance on water movement in the soil-root-leaf pathway (Meinzer et al., 1995), which can be increased by waterlogging (Kubota et al., 2023), a condition frequently observed in our study area.

Meinzer, F. C., Goldstein, G., Jackson, P., Holbrook, N. M., Gutierrez, M. V., & Cavelier, J. (1995). Environmental and physiological regulation of transpiration in tropical forest gap species: the influence of boundary layer and hydraulic properties. Oecologia, 101, 514-522.

Kubota, S., Nishida, K., & Yoshida, S. (2023). Plant hydraulic resistance controls transpiration of soybean in rotational paddy fields under humid climates. Paddy and Water Environment, 21(2), 219-230.

8. The statement on line 333-334 is too strong and even might be wrong. This statement only considers the biological impact (transpiration) but ignores the physical impact (E) of VPD. The similar to the effect of surface conductance. Need to be rephrased.
R: We think this question is related to the previous one. As explained above, we rephrased the text.

9. The whole discussion section is more like 'Result'. Please put some effort on what are the implication of the analysis and what are the biological meanings of the diagnostics (especially the decoupling of surface conductance and stomatal conductance, the different expression of WUEs).
R:  Based on this comment, we modified several sections of the Discussion. The lines in the new document are: 295-298, 303-304, 312-317, 319-324, 334-340, 358-359, 366-367, 379-386,

10. Since the structural equation model is more often used in the social-economics, I would like to see detailed description of this model and what the potential uncertainties in using this model in this work, especially the potential issues of multicollinearity (occurs when the independent variables are highly correlated, which can inflate the standard errors and reduce the statistical significance of the estimates) and endogeneity (occurs when the independent variables are correlated with the error term, which can bias the estimates and invalidate the causal inference in the model).

R: Thank you for the question. Please refer to question 7 regarding variable collinearity. PLS-PM was indeed created and used initially in socioeconomics but has also been used in ecological (e.g., Fernandes et al., 2019) and climatic applications (e.g., Yu and Leng, 2022). The main difference between PLS-PM and likelihood-based SEM (covariance-based SEM; CBSEM), for example, is that it uses ordinary least square regressions for estimating the path coefficients. However, another important difference is that PLS-PM is flexible to model interactions using a reflective or a formative conceptualization, which greatly alters the way of the measurement approximation (e.g., effect indicator, causal indicator, or composite indicator). This approach is different from common factor-based SEM, which considers the constructs as common factors that explain the covariation between their associated indicators. Sarstedt et al. (2016) found through simulations that PLS-PM and SEM do not differ significantly when reflective modes are used (the default for the PLS-PM and the only option for SEM). They also found that PLS entails practically no bias when estimating data

from a composite model population, regard- less of whether the measurement models are reflective or formative, while SEM showed biases when using composites in formative models. Hence, PLS-PM is often robust if one is not sure of the nature of the data and when there is uncertainty about whether to use formative or reflective models. We added this information as (lines 202-208):

"PLS-PM uses ordinary least square regressions for estimating the path coefficients and has been found to be flexible to model interactions using a reflective or a formative conceptualization, which dramatically alters the way of the measurement approximation (e.g., effect indicator, causal indicator, or composite indicator). Through simulations, Sarstedt et al. (2016) found that PLS-PM and factor-based SEM do not differ significantly when reflective modes are used (the default for the PLS-PM and the only option for SEM). They also found that PLS entails practically no bias when estimating data from a composite model population, regardless of whether the measurement models are reflective or formative. Hence, PLS-PM is often robust if one is unsure of the nature of the data".

However, PLS-PM does not allow correlation with the error terms as other covariance-based methods, which is one of their advantages.

Fernandes, A. C. P., Fernandes, L. S., Moura, J. P., Cortes, R. M. V., & Pacheco, F. A. L. (2019). A structural equation model to predict macroinvertebrate-based ecological status in catchments influenced by anthropogenic pressures. Science of the Total Environment, 681, 242-257.

Sarstedt, M., Hair, J. F., Ringle, C. M., Thiele, K. O., & Gudergan, S. P. (2016). Estimation issues with PLS and CBSEM: Where the bias lies! *Journal of Business Research*, *69*(10), 3998–4010.

Yu, L., & Leng, G. (2022). Identifying the paths and contributions of climate impacts on the variation in land surface albedo over the Arctic. Agricultural and Forest Meteorology, 313, 108772.

Minor Comments:
1. There is no need to abbreviate carbon to 'C'.
R: We followed the recommendation and removed this abbreviation

2. Typo on line 39-40.
R: All typos related to references were fixed.

3. Line 77, need to add 'Leaf Area Index'.
R: Added

4. Line 86, Fig. 1e.

R: Hyphen removed

5. Line 123, need a citation for the MDS
R: Done

6. line 127, great to see this important point, do you also incorporate the heat storage to H and LE?
R: We did not incorporate the heat storage into the analysis. We assumed that all the energy stored in the canopy structure, the soil water, and the litter was the primary cause of the missing energy at the peatland site and a good part of the missing energy in the forest, the latter possibly also being affected by synoptic-scale transport.
To better estimate LE (and therefore ET) we tackled this issue with the correction proposed by Mauder et al (2013), so the energy balance residue, evaluated daily, was used to partition that energy into LE and H (preserving the Bowen ratio) to incorporate it into the total energy measured.
Now we have included a paragraph that explains this in the paper (lines 127-140).

"The EBR for both sites was 0.7 and 0.93 for the forest and peatland ecosystems, respectively. The missing energy in both the peatland and forest ecosystems could be connected to the energy used spent by photosynthesis, and the energy stored in the canopy structure, the soil water, and the litter. The lack of closure could also be understood as a systematic underestimation of H and LE related to synoptic-scale transport phenomena that are not captured by the EC systems (Mauder et al, 2013). There is a larger discrepancy in energy balance closure with the forest as the magnitude of synoptic scale transport is likely to be larger in taller canopies and as the forest's canopy structure can store more energy because of a much larger biomass than the other ecosystem measured. Although this issue in the forest could be approached by having a longer averaging period, e.g. 60 minutes instead of 30 minutes, increasing the time average leads to other problems related to non-stationarity (Mauder and Foken, 2006). Based on these assumptions, we implemented the Mauder et al. (2013) correction which uses the energy balance residues, evaluated daily, to partition that energy between H and LE in a way that preserves the Bowen ratio."

7. line 141, what does 'valid' mean? Good quality data?
R: Valid data in this case are those that met the four criteria described in the previous paragraph.

8. please check equation 3 (should be uW U Ep) and equation 4
R: Thanks for catching this error. The correct equation is T=(uWUEa/uWUEp) ET.

9. line 150, from the uWUE paper by Zhou et. al. (2016), the procedure should be done for each site-year not the entire site. So you compute the uWUEp based on all data in the site?
R: Thank you for your comment. The annual values of uWUEp in Zhou et al., 2016 are only calculated to validate their assumption of uWUEp being constant in each site.

In section 3.1 of Zhou et al 2016, the annual uWUEp values were expressed in terms of their departures from the long-term mean in percent to validate the sites that drift less than 10%. Our sites show 8.5% and 3.3% departures from the means for the forest and peatland, respectively. We have now included a paragraph in the paper for a more detailed explanation (lines 172-174).

"As in Zhou et al., 2016, uWUEp is here assumed to be constant for each site and representative of it when it is calculated as a long-term average uWUEp. This can be assured because the standard deviation of the annual values of uWUEp are 8.5% and 3.3% departed from long-term averages of uWUEp in the forest and peatland, respectively."

10. line 159, consider to remove and it's already mentioned.
R: Sentence removed.

11. why do you indentify the drives at hourly scale not daily scale? how do you deal with nightime data?
R: In section 2.4 we describe 4 conditions that the data have to meet, including that we used only daytime data. To clarify this, we modified the first sentence in section 2.6.

12. the nRMSE is somehow wrong here. what is the 1/2 in the numerator? should not it be 1/n?
R: We thank the reviewer for noticing this mistake. Indeed the 1/2 in the numerator should be a 1/n. We have corrected equation 8:

$$nRMSE = \left( \frac{\sqrt{\frac{1}{n} \sum_{i=1}^{n} (y_j - \hat{y}_j)^2}}{max(X) - min(X)} \right) \times 100$$

13. Figure 2a, why did the albedo change obviously after 2018 and the change is corresponding to the change in surface temperature?
R: Although we agree that from the figure it seems that albedo increased in the peatland after 2018 and that temperature seems to decrease accordingly, due to the limited number of years of data we could not test for the significance of any trend.

14. line 241, 'Fig 5' to 'Figure 5' as its the start of the paragraph.
R: It is now rephased as:
"The correlation matrices for carbon and water fluxes and WUE are shown in Fig 5."

15. Figure 5. From desecration of the sites (high Precip and low T/ET,ET/P), these two sites are energy limited and this explains the higher correlation between ET and E than T, as well as low. But I do not see why does WUE have a higher relationship with ET instead of GPP given the fact that water is not a limiting factor. This is not discussed in the discussion section.

R: we modified the text to explain why WUE has a higher relationship with ET instead of GPP (lines 358-363).

"The large contribution that E makes to ET (Figure 6) may be explained by the frequently waterlogged and humid conditions in this area, where water can evaporate directly from soils and wet vegetated surfaces This, together with the presence of plant species that either do not have stomata (i.e., mosses in the peatland or epiphytes in the forest) or, in the forest, have poor stomatal control results in a decoupling of WUE from transpiration and GPP. A previous study showed that one of the dominant species in the forest (*Drimys winteri*) has traits that are focused more on efficient water transport favoring carbon gain over the ability to regulate water loss (Negret et al., 2013)."

The authors have addressed most of my previous concerns, and I appreciate their efforts. However, there are two parts that are still missing in the draft:
- I do not think the authors have addressed my comments regarding the energy balance closure problem, at least they can check the relative humidity dependent water flux underestimates (i.e. plot LE/(Rn-G-H) as a function of relative humidity). If the problem is there, I think the authors need to do a related correction (the eddy pro can not solve this problem). Such a problem has been reported for most of sites in the eddy-tower network.

We appreciate your review and constructive feedback. In response to your concern about the energy balance closure problem, we want to emphasize that we carefully considered the issue and applied corrections to address potential underestimates in latent heat flux.

Specifically, we implemented the High Relative Humidity Correction (HRHC) described in Zhang et al. (2023) method to account for humidity-dependent flux underestimations. Prior to the application of any correction, a substantial ~30% energy deficit was observed in the forest site, highlighting a significant gap in the energy balance ratio (EBR). Although the HRHC method slightly improved the closure by increasing latent heat flux by 6.1% in SDP and 2.5% in SDF, the overall deficit persisted. Moreover, the resulting ET in the peatland was higher compared to the forest, which we think does not make biological sense. Considering this, we opted for keeping the Bowen Ratio Correction (BRC; Mauder et al., 2013) as a more robust correction approach, acknowledging the potential risk of overestimating evapotranspiration using the latter method.

| Increase Ratio | RH = 40% | RH = 50% | RH = 60% | RH = 70% | RH = 80% | RH = 90% |
|---|---|---|---|---|---|---|
| LEcorr / LE in Forest | 2.5 % | 2.6 % | 3.0 % | 4.1 % | 7.2 % | 21.5 % |
| LEcorr / LE in Peatland | 6.2 % | 6.5 % | 7.8 % | 11.4 % | 18.2 % | 36.4 % |

To clarify this, we added the following text to the Methods section (lines 140-158):

"Additionally, high relative humidity can produce an underestimation of LE, especially with closed-path systems, as the cut-off frequency of the closed-path system for water vapor concentration measurements decreases exponentially with increasing relative humidity (Zhang et al., 2023a).

Based on these assumptions, we implemented two corrections separately:

1) the Mauder et al. (2013) correction, hereafter the Bowen Ratio Correction (BRC), which uses the energy balance residual, evaluated on a daily basis, to partition the residual between H and LE in a way that preserves the Bowen ratio.
2) the Zhang et al. (2023a) correction, hereafter the High Relative Humidity Correction (HRHC), which rectifies LE considering the impact of high relative humidity.

Prior to the application of HRHC and BRC, a substantial 30% energy deficit was observed in the forest site. After the application of the HRHC, LE increased by only 6.1% in the

peatland and 2.5% in the forest, while the calculated ET using HRHC was smaller in the forest than in the peatland, contrary to expectations based on their canopy leaf areas. Due to the observed limitations in the ability of HRHC to accurately capture LE variations in cases of poor EBR, we opted for the BRC over the HRHC.

Despite the acknowledged risk of potentially overestimating evapotranspiration using EBR, it provided a more robust correction approach compared to HRHC when comparing ET in both ecosystems. Furthermore, the decision to exclusively apply the Bowen Ratio Correction (BRC) was influenced by the challenge of simultaneously using both corrections, as they operate on different principles and may introduce complexities in interpreting the corrected results. Nonetheless, we report the estimation of ET using both corrections (Supplementary material, Tables S2) and their partitioning values (Supplementary material, Tables S3 and S4)."

**Table S2.** Evapotranspiration in the forest (SDF) and peatland (SDP) using the High Relative Humidity Correction (HRHC) and the Bowen ratio correction (BRC).

| Evapotranspiration $[mm\ year^{-1}]$ | SDF using HRHC | SDF using BRC | SDP using HRHC | SDP using BRC |
|---|---|---|---|---|
| 2015 | 498 | 745 | 619 | 584 |
| 2016 | 587 | 820 | 615 | 598 |
| 2017 | 736 | 1122 | 697 | 738 |
| 2018 | 703 | 1089 | 794 | 763 |
| 2019 | 690 | 1071 | 701 | 651 |
| 2020 | 594 | 936 | 689 | 770 |
| 2021 | 492 | 721 | 629 | 670 |
| 2022 | 508 | 772 | 640 | 681 |
| Mean ± SE | 601 ± 35 | 910 ± 59 | 673 ± 21 | 682 ± 25 |

- Secondly, I have an additional suggestion that the authors can double-think to provide to better support the conclusions drawn from Figures 6, 7 and 8 (This is an diagnostic test in this part but I highly suggest to do so). The use of a single ET partitioning method is not robust enough to draw these conclusions in such a knowledge-limited region, one might partition ET using a machine-learning-based method (e.g., the partitioning method proposed by Nelson et al., 2018), where you can analyse the importance of the variables at the mean time, This would give the
reader more confidence in the ET partitioning and related arguments.

Thank you for your suggestion and support. We calculated the partitioning of ET using the methodologies proposed by Zhou et al. (2016) and Nelson et al. (2018), using data from two different corrections: the Bowen ratio correction (BRC) and the High Relative

Humidity Correction (HRHC) methods. Based on the results, we found that the partitioning proposed by Nelson yields an even lower contribution of T to ET compared to the Zhou partitioning method. We think this supports the results of the ET partitioning and related arguments.

To clarify this point, we added text in the Discussion section (line 390-395):

"We evaluated the partitioning method proposed by Nelson et al. (2018) to compare it with the method used in our work (Zhou et al., 2016) and found that the former method yielded an even higher contribution of evaporation to ET. The results are shown in the Supplementary material, Tables S3 and S4. Furthermore, we found that the relationships between biometeorological variables and evaporation and transpiration fluxes were consistent between both methods (data not shown). While acknowledging the potential for variations and complexities when applying different partitioning methods in natural ecosystems, we think this supports our results."

**Table S3.** Contribution of transpiration (T) to evapotranspiration (ET) in the forest site (SDF) using the High Relative Humidity Correction (HRHC) and the Bowen ratio correction (BRC), associated with the partitioning methods proposed by Zhou et al. (2016) and Nelson et al. (2018).

| T/ET [%] | Zhou (2016) using HRHC | Zhou (2016) using BRC | Nelson (2018) using HRHC | Nelson (2018) using BRC |
|---|---|---|---|---|
| 2015 | 60.6 | 55.8 | 41.5 | 37.5 |
| 2016 | 52.1 | 49.0 | 44.3 | 38.5 |
| 2017 | 47.0 | 40.1 | 39.9 | 35.6 |
| 2018 | 49.3 | 42.6 | 40.3 | 36.6 |
| 2019 | 46.3 | 39.7 | 41.8 | 34.5 |
| 2020 | 51.6 | 45.2 | 40.5 | 34.3 |
| 2021 | 57.0 | 51.3 | 40.4 | 38.4 |
| 2022 | 49.7 | 44.3 | 37.3 | 36.2 |
| Mean ± SE | 51.7 ± 1.7 | 46 ± 2 | 40.8 ± 0.7 | 36.5 ± 0.6 |

**Table S4.** Contribution of transpiration (T) to evapotranspiration (ET) in the peatland site (SDP) using the High Relative Humidity Correction (HRHC) and the Bowen ratio correction (BRC), associated with the partitioning methods proposed by Zhou et al. (2016) and Nelson et al. (2018).

| T/ET [%] | Zhou et al 2016 using HRHC | Zhou et al 2016 using BRC | Nelson 2018 using HRHC | Nelson 2018 using BRC |
|---|---|---|---|---|
| 2015 | 51.6 | 48.2 | 43.5 | 37.5 |
| 2016 | 56.1 | 52.8 | 49.0 | 42.7 |
| 2017 | 54.1 | 47.5 | 48.1 | 37.8 |
| 2018 | 46.9 | 46.7 | 44.9 | 34.3 |
| 2019 | 51.5 | 51.2 | 47.3 | 35.7 |
| 2020 | 40.8 | 37.3 | 37.0 | 29.1 |
| 2021 | 57.1 | 52.6 | 52.4 | 41.2 |
| 2022 | 57.0 | 52.5 | 55.3 | 39.1 |
| Mean ± SE | 51.9 ± 2.0 | 48.6 ± 1.8 | 47.2 ± 2.0 | 37.2 ± 1.5 |